Adaptive genetic variation at three loci in South African vervet monkeys (Chlorocebus pygerythrus) and the role of selection within primates

http://orcid.org/0000-0003-2189-5539 Coetzer Willem G. 1 coetzerwg@ufs.ac.za
Turner Trudy R. 2
Schmitt Christopher A. 3
Grobler J. Paul 1
1 Department of Genetics, University of the Free State , Bloemfontein , South Africa
2 Department of Anthropology, University of Wisconsin–Milwaukee , Milwaukee , WI, USA
3 Department of Anthropology, Boston University , Boston , MA, USA
Hoover Kara
Electronic publication date: 2018 Jun 4
Publication date: 2018
Volume: 6
Electronic Location ID: e4953
Received 2018 Feb 23; Accepted 2018 May 22
Copyright: © 2018 Coetzer et al.
Copyright year: 2018
Copyright holder: Coetzer et al.
License: This is an open access article distributed under the terms of the Creative Commons Attribution License, which permits unrestricted use, distribution, reproduction and adaptation in any medium and for any purpose provided that it is properly attributed. For attribution, the original author(s), title, publication source (PeerJ) and either DOI or URL of the article must be cited.
License URL: https://creativecommons.org/licenses/by/4.0/

Keywords: Pathogen diversity, Environmental factors, Adaptive variation, Vervet monkey

Funding: University of the Free State Postdoctoral Fellowship Grant The study was funded by the University of the Free State Postdoctoral Fellowship Grant. The funders had no role in study design, data collection and analysis, decision to publish, or preparation of the manuscript.

==============================
Vervet monkeys (Chlorocebus pygerythrus) are one of the most widely distributed non-human primate species found in South Africa. They occur across all the South African provinces, inhabiting a large variety of habitats. These habitats vary sufficiently that it can be assumed that various factors such as pathogen diversity could influence populations in different ways. In turn, these factors could lead to varied levels of selection at specific fitness linked loci. The Toll-like receptor (TLR) gene family, which play an integral role in vertebrate innate immunity, is a group of fitness linked loci which has been the focus of much research. In this study, we assessed the level of genetic variation at partial sequences of two TLR loci (TLR4 and 7) and a reproductively linked gene, acrosin (ACR), across the different habitat types within the vervet monkey distribution range. Gene variation and selection estimates were also made among 11–21 primate species. Low levels of genetic variation for all three gene regions were observed within vervet monkeys, with only two polymorphic sites identified for TLR4, three sites for TLR7 and one site for ACR. TLR7 variation was positively correlated with high mean annual rainfall, which was linked to increased pathogen abundance. The observed genetic variation at TLR4 might have been influenced by numerous factors including pathogens and climatic conditions. The ACR exonic regions showed no variation in vervet monkeys, which could point to the occurrence of a selective sweep. The TLR4 and TLR7 results for the among primate analyses was mostly in line with previous studies, indicating a higher rate of evolution for TLR4. Within primates, ACR coding regions also showed signs of positive selection, which was congruent with previous reports on mammals. Important additional information to the already existing vervet monkey knowledge base was gained from this study, which can guide future research projects on this highly researched taxon as well as help conservation agencies with future management planning involving possible translocations of this species.

Introduction

Data on the level of genetic adaptation within animal populations is an important aspect of conservation biology. Population or habitat specific adaptations should be considered when studying animal population dynamics and evolution, as these aspects of population genetic diversity will undoubtedly provide important information to guide conservation management (Funk et al., 2012). Neutral genetic variation has long been the marker of choice but can overestimate the amount of genetic diversity within a population and therefore overestimate a population’s viability. The consideration of both adaptively linked and neutral markers during population level studies is therefore important (Funk et al., 2012; Hartmann, Schaefer & Segelbacher, 2014).

Vervet monkeys (Chlorocebus pygerythrus) are one of the most widely distributed non-human primate species found in South Africa, with only Papio ursinus found in a wider range (Hoffmann & Hilton-Taylor, 2008). This species occurs across all the South African provinces, inhabiting a large variety of habitats (Skinner & Chimimba, 2005; Kingdon et al., 2008), including xeric and mesic zones. These primates are listed as ‘Least concern’ in the Red list of mammals of South Africa, Swaziland and Lesotho (Turner et al., 2016a). The genus Chlorocebus is a taxon of significant scientific interest within research areas such as immunodeficiency virus pathology (Ma et al., 2013), parasite ecology (Gaetano et al., 2014; Wren et al., 2015; Chapman et al., 2016), neuroscience (Woods et al., 2011; Mendell et al., 2014), social learning (Botting et al., 2018) and sexual selection (Borgeaud et al., 2015; Rodríguez et al., 2015), and these studies require a clear understanding of genetic boundaries and connectivity between populations, using both neutral and adaptive markers. Vervet monkeys also often find their way to primate rehabilitation centres following human primate conflicts (Wimberger, Downs & Boyes, 2010) and re-introduction of these animals to suitable recipient populations without considering the provenance of the rehabilitated animals should be discouraged. A recent study focusing on South African vervet monkey phylogeography identified mtDNA structuring among populations linked to current and past habitat distributions, geographic barriers, distance and female philopatry (Turner et al., 2016b). Three broad genetic clusters were identified, corresponding to (i) the northern part of the distribution range, including the northern part of the Indian Ocean coastal belt, (ii) the central regions of South Africa and (iii) the southern part of the Indian Ocean coastal belt and adjacent inland areas. Turner et al. (2016b) urged that the addition of nuclear loci should be considered in future research to provide a better understanding of the influence of selection on the observed genetic structuring seen among South African vervet monkey populations. A genome wide study by Svardal et al. (2017) showed that viruses played an important role during Chlorocebus evolution, which is in line with the findings of Enard et al. (2016) who identified viruses as the main driver of evolutionary adaptation in mammals. Strong signals for selection were specifically identified for genes involved in cell signalling and transcriptional regulation following viral exposure in Chlorocebus monkeys (Svardal et al., 2017). Further research into the adaptive genetic diversity of vervet monkeys will not only be of value for future studies on this widely researched taxon but can also provide conservation authorities with the needed information to make informed decisions with regards to possible translocations.

The different habitats within the vervet distribution range differ sufficiently that it can be assumed that different environmental factors could drive adaptation in local populations. This could then lead to selection at specific fitness linked loci. In recent years, there has been an increasing trend to use adaptive variation in population and phylogenetic analyses (Vasemägi & Primmer, 2005; Gatesy & Swanson, 2007; Ayoub et al., 2009; Gonzalez-Quevedo et al., 2015; Quéméré et al., 2015). Fitness-linked genes are influenced by external elements such as host-pathogen interactions (Vasemägi & Primmer, 2005; Holderegger, Kamm & Gugerli, 2006), which directly affect population fitness. The study of variation at adaptive loci can therefore provide valuable information on interaction between environment, genotype and the selective influences that shape patterns of diversity.

Pathogen-mediated selection is widely accepted as an important evolutionary driver in nature (Little, 2002). In this regard, the Toll-like receptor (TLR) gene family, which play an integral role in vertebrate innate immunity, is a group of fitness linked loci which has been the focus of much phylogeographic and phylogenetic research (Wlasiuk & Nachman, 2010; Tschirren et al., 2012; Hartmann, Schaefer & Segelbacher, 2014). TLRs respond to a wide range of pathogen-associated molecular patterns (PAMPs) (exogenous and endogenous, Roach et al., 2005); for example, lipoteichoic acids (gram-positive bacteria, Takeuchi et al., 1999), lipopolysaccharides (gram-negative bacteria, Poltorak et al., 2000), CpG DNA (bacterial, Hemmi et al., 2000; viral, Lund et al., 2003) and single-stranded RNA (viral, Diebold et al., 2004; Lund et al., 2004) to name a few. There are generally two classes of TLRs; one recognising mainly extracellular threats such as lipopolysaccharides and expressed on the outer cell membrane, and the other being expressed within intracellular organelles and recognising mainly intracellular threats such as CpG DNA (Takeda, Kaisho & Akira, 2003; Roach et al., 2005; Vinkler & Albrecht, 2009; Fornůsková et al., 2013). Two TLR genes were selected for this study on the basis of the PAMPs associated with each. First, TLR4 was selected due to its responds to a range of PAMPs including lipopolysaccharide from gram-negative bacteria as well as the fusion protein of respiratory syncytial virus (Kurt-Jones et al., 2000), whereas TLR7 was selected due to its involved in immune responses to viral ssRNA (Diebold et al., 2004; Lund et al., 2004) and phagosomal bacteria (Mancuso et al., 2009). These two TLR genes covers a wide range of possible pathogen-mediated responses. Gram-negative bacteria can cause various infections, from pathogenic Escherichia coli causing gastrointestinal infections (Manning, 2010) to Klebsiella pneumoniae causing pneumonia (Aujla et al., 2008; Twenhafel et al., 2008) and Neisseria meningitidis being a causal agent of meningococcal meningitis (Brandtzaeg, 2006).

Non-human primates are also affected by numerous viral pathogens. African green monkey simian immunodeficiency virus (SIVagm) is a ssRNA lentivirus (Kaup et al., 2005) associated with primates from the genus Chlorocebus. These viruses do, however, not cause AIDS symptoms in their host species and is therefore studied at length to identify the reasons for this lack of symptoms (Ma et al., 2013). Other ssRNA viruses related to Chlorocebus species include simian hepatitis A virus (Balayan, 1992), influenza (Webster et al., 1992) and a range of simian arteriviruses and simian pegiviruses (Bailey et al., 2014a, 2014b, 2016). Using these two genes would therefore provide valuable information with regards to vervet monkey adaptations to viral and bacterial pathogens.

The use of a reproductively linked gene can provide additional information with regards to local adaptations among populations, especially long separated populations. The acrosin (ACR) gene encodes for the serine proteinase ACR, which is linked to fertility in animals and thought to be involved in secondary binding of the sperm cell to the egg zona pellucida leading to proteolysis of the extracellular egg coat (Keime, Adham & Engel, 1990; Gatesy & Swanson, 2007). Reports indicated that this gene show significant signs of adaptive change within mammals (Swanson et al., 2001), making it a good candidate for the assessment of the influences of population isolation within a species.

Here, we assessed the level of genetic variation at partial sequences of one reproductively-linked gene (ACR) and two immune-linked loci (TLR4 and TLR7) across eight habitat types within the vervet monkey distribution range, to complement the work by Turner et al. (2016b) using a neutral mtDNA locus. We also investigated the effect of natural selection on the evolution of these gene segments within a wider sample of primates. This study will add substantially to our current knowledge of vervet monkey evolution in South Africa, which could aid future research projects as well as conservation management plans.

Methods

Ethics

Sampling approval and research ethical clearance was obtained for a previous study using mtDNA analysis (Turner et al., 2016b) from the Institutional Care and Use Committee of the University of Wisconsin–Milwaukee (Ref no. 07-08 #32) and the Inter-Faculty Animal Ethics Committee of the University of the Free State, South Africa (Ref no. UFS-AED13/2010). All samples from this study were sourced from the Turner et al. (2016b) study and no new additional samples were included.

Samples and DNA extraction

Samples from a total of 81 vervet monkeys were included in this study (Table S1), representing nine localities from across the distribution range (Turner et al., 2016b), located in five biomes (Nama-Karoo, Savanna, Grassland, Albany thicket and Indian Ocean coastal belt biomes; Table 1; Fig. 1A) and eight bioregions/vegetation types. These localities represent significantly different environmental conditions, based on parameters such as the mean annual precipitation (MAP), mean annual ambient temperature (MAT) and maximum annual temperature (MaxT) (Table 1). Due to the philopatric nature of vervet monkey troops, there is expected to be a certain level of relatedness among members of the same troop. We therefore aimed to select specimens from multiple troops, if possible. All samples were sourced from the Department of Genetics, University of the Free State, Biobank and were previously collected by Turner et al. (2016b). All DNA extracts were obtained from ear biopsies. DNA extraction was performed using the Roche High Pure polymerase chain reaction (PCR) Template Preparation Kit (Roche Diagnostics, Indianapolis, IN, USA). Sample preparation included a re-hydration step, followed by the manufacturer’s instructions. The initial digestion step was carried out overnight. All DNA samples were stored at −20 °C.

Table 1 Sampling sites of vervet monkeys within the South African distribution range.

Locality	Site	Geographic locality	Sample size	Biome	Bioregion unit/vegetation type	Average annual precipitation (mm)	Average annual ambient temperature (°C)	Maximum annual temperature (°C)	
Northern	N1	Central Limpopo province	11	Savanna	Polokwane plateau Bushveld	500	16.9	33.2	
North-eastern	NE1	Mpumalanga Lowveld	10	Savanna	Granite Lowveld	633	20.9	38.75	
North-western	NW1	Orange river, Northern Cape	10	Namma-Karoo	Lower Gariep alluvial vegetation	131	20.5	39.5	
Central	C1	North-western Free State	10	Savanna	Kimberley Thornveld	407	17.2	37.45	
C2	Central Free State	10	Grassland	Highveld alluvial vegetation	495	16.6	32*	
Southern coastal belt	SC1	Shamwari, Eastern Cape	10	Albany thicket	Kowie thicket	517	17.5	35	
Northern coastal belt	NC1	St. Lucia, KwaZulu-Natal	10	Indian Ocean coastal belt	Northern coastal forest	1,044	21	35.3	
NC2	Thorny Park, KwaZulu-Natal	5	Indian Ocean coastal belt	KwaZulu-Natal coastal belt	989	19.6	32.6	
NC3	Blythedale, KwaZulu-Natal	5	Indian Ocean coastal belt	KwaZulu-Natal coastal belt	989	19.6	32.6	
Notes:

General geographic locality, average annual precipitation, average ambient temperature and maximum annual temperature are also provided. Site and geographic locality information follows that from Turner et al. (2016b). The environmental data were taken from Mucina & Rutherford (2006), unless otherwise stipulated. The provided environmental data are the values provided for the specific Bioregion unit or Veld type.

* Average maximum temperature observed for Bloemfontein over the last nine years, taken from http://www.worldweatheronline.com.

Figure 1 South African maps providing sampling locality information and haplotype frequencies for the two TLR loci.

(A) Biome map indicating vervet monkey sampling localities across different biomes. The vervet monkey distribution range is indicated by the grey shading. The software DIVA-GIS (http://www.diva-gis.org/) was used to create the vervet distribution range and biome map, which is based on information obtained from Hoffmann & Hilton-Taylor (2008) and Mucina & Rutherford (2006). (B) Vervet TLR4 haplotype frequencies per locality and (C) vervet TLR7 haplotype frequencies per locality.

DNA amplification and sequencing

Primers for amplification of partial segments from the ACR, TLR4 and TLR7 genes were designed using Primer3 (Rozen & Skaletsky, 1999) as implemented in Geneious v9 software (Kearse et al., 2012). Primers were designed using the online GenBank genome sequences for Chlorocebus sabaeus (ACR, Chromosome 19: NC_023660; TLR4, Chromosome 12: NC_023653; TLR7, Chromosome X: NC_023671.1). See Table 2 for the primer sequences and annealing temperature information. The ACR amplicons covered a partial segment of exon 4 and the complete intron 4 (Chl_ACR2) and complete exon 5 (Chl_ACR1), totalling a ∼1,000 bp sequence. This region was selected because it was reported that exon 5 is the most rapidly evolving segment of the mammalian ACR gene (Gatesy & Swanson, 2007). The ACR intronic region was a by-product from sequencing the ACR exon 4 and exon 5 regions. The TLR4 primers covered a partial ∼700 bp segment of exon 3, with the TLR7 primers amplifying a partial ∼1,350 bp segment of exon 3. The first ∼700 bp of the TLR4 exon 3 gene was selected as it is the largest of the three TLR4 exons observed in primates and it contains the extracellular leucine-rich repeat domain, which is responsible for ligand recognition (Janssens & Beyaert, 2003). The TLR7 protein is mainly encoded by the third exon of this gene (Astakhova et al., 2009). The region targeted in our study covers approximately a third of the 5′ end of exon 3, which also contains the extracellular domain. Two PCR kits (KAPA HiFi HotStart ReadyMix PCR kit; KAPA Biosystems, Cape Town, South Africa and Ampliqon TEMPase Hot Start 2x Master Mix, Odense, Denmark) were used, due to product availability. The manufacturer protocols were followed for both kits. The annealing temperatures (Ta) for the different gene segments are provided in Table 2.

Table 2 Primer sequences and annealing temperatures for the three gene regions amplified in this study.

Primer	Forward sequence (5′-3′)	Reverse sequence (5′-3′)	KAPA HiFi Ta	Ampliqon TEMPase Ta	
Chl_ACR1	CAGCAGGAAACCATGTGACT	TTTTCTCAGCACTGAAGGGC	–	64 °C	
Chl_ACR2	TATGCTGATGGAGGCACGTGTGGA	GCGCCTTTCTTGCTGTCTTTGCAC	–	64 °C	
Chl_TLR4	ACAGAAGCTGGTGGCTGTGG	TTGAAAGCAACTCTGGTGTG	66 °C	64 °C	
Chl_TLR7a	ACCATGTGATCGTGGACTGC	GGGGCACATGCTGAAGAGAG	66 °C	64 °C	
Chl_TLR7b	TGCCCCATTTCCTTGTACGC	AAGCCGGTTGTTGGAGAAGTC	62 °C	64 °C	
Note:

The ACR primer pair was only amplified with the Ampliqon TEMPase Hot Start 2x Master Mix. All primer sequences are provided in 5′ to 3′ direction.

Amplification success was assessed on a 1% Agarose gel. PCR amplicons were purified using the BioSpin PCR Purification Kit (BioFlux, Tokyo, Japan). The ABI PRISM® BigDye® Terminator v3.1 Cycle Sequencing Kit (Applied Biosystems Division, Foster City, CA, USA) was used for all sequencing reactions. Sequences were analysed with an ABI 3130 Genetic Analyzer, following sequence product clean-up with the ZR DNA Sequencing Clean-up™ Kit (Zymo Research, Irvine, CA, USA).

Sequence diversity within vervet monkeys

All raw sequences were assembled and edited using Geneious. Sequence alignments for each gene region were performed with ClustalW (Thompson, Higgins & Gibson, 1994) as implemented in Geneious. Diploid genotypes were resolved into haplotypes using the Bayesian method in Phase v2.1 (Stephens, Smith & Donnelly, 2001; Stephens & Scheet, 2005) implemented in DnaSP v5.10.01 (Librado & Rozas, 2009). TLR7 is located on the X chromosome, and we therefore only performed haplotype inference analysis for the sequences from females, as only one gene copy is present in males. The hemizygous male sequences were then added to the phased female sequences for downstream analyses. The number of iterations were set to 50,000, with a burn-in of 5,000, and with all other settings at default values. Samples were grouped according to eight sampling regions for all analyses. Data from NC2 and NC3 were combined due to small sample sizes, the close proximity between sites and similar environmental conditions observed between these sites (Table 1). Summary statistics performed with DnaSP included the number of polymorphic sites (S), number of non-synonymous sites, number of haplotypes (h), haplotype diversity (Hd) and nucleotide diversity (π, rounded to four decimal places due to small magnitude). The program PopArt (Leigh & Bryant, 2015) was used to assess the relationship between haplotypes by generating a minimum spanning network of each gene region.

Protein homology modelling was used to provide a clearer perspective of the position of each partial sequence used in the current study. The partial sequences were translated to amino acid sequences using Geneious software. Homology modelling was then performed in SWISS-MODEL (Biasini et al., 2014), and the partial protein structures were superimposed on reference protein structures using DeepView/Swiss-PdbViewer (Guex & Peitsch, 1997; http://www.expasy.org/spdbv/). No complete Chlorocebus ACR mRNA or amino acid sequences were available for comparison. We therefore used SWISS-MODEL to identify the best fit template for model building. An ACR protein structure for Ovis aries beta-acrosin (PDB ID: 1fiw; Tranter et al., 2000) was identified as the best fit. Our vervet partial ACR sequence was then superimposed onto this structure using DeepView/Swiss-PdbViewer (Guex & Peitsch, 1997; http://www.expasy.org/spdbv/). A Chlorocebus TLR4 protein structure was modelled for use as a reference structure using a Chlorocebus sabaeus amino acid sequence downloaded from the Uniprot Knowledgebase (UniProtKB accession: A0A0D9RL22), with a published human TLR4 protein reference structure (PDB ID: 4G8G; Ohto et al., 2012). The partial sequences generated from our study was then modelled according to, and then superimposed onto, this Chlorocebus sabaeus TLR4 protein structure. A reference protein structure for Chlorocebus TLR7 was constructed using a Chlorocebus sabaeus amino acid sequence from the Uniprot Knowledgebase (UniProtKB accession: A0A0D9SDT9) and a Macaca mulatta protein structure as a reference (PDB ID: 5gmf; Zhang et al., 2016). Our partial Chlorocebus pygerythrus TLR7 sequences were then superimposed on this Chlorocebus sabaeus scaffold.

A Mantel test was performed in Arlequin v3.5 (Excoffier & Lischer, 2010) to estimate the correlation between geographic distance and the observed genetic diversity. The pairwise FST values generated in Arlequin were used as measure of genetic distance, with the geographic distance calculated using an online tool for latitude/longitude to geographic distance conversions (Veness, 2017). The correlation between the estimated Hd against each of three environmental factors, MAP, MAT and MaxT, were assessed via generalized linear model (GLM) analysis, using PAST v3 software (Hammer, Harper & Ryan, 2001). The environmental data were taken from Mucina & Rutherford (2006), which is the most widely used and complete vegetation and biome data for South Africa. A GLM analysis was used as our Hd data did not conforming to normality. PAST implements the basic version of the GLM, and a least squares linear regression was utilised due to its robustness. Correlation analyses using Hd values calculated from only the non-synonymous sites were also performed. A correlation analysis was also conducted using the Hd values for the mitochondrial D-loop region assessed by Turner et al. (2016b) and the MAP, MAT and MaxT values for the corresponding regions assessed in the current study. This assessment was done to compare neutral genetic variation patterns to that observed from adaptive markers.

Assessment of natural selection within South African vervet monkeys

The TLR4 and TLR7 sequence alignments for the South African vervet monkeys were used to assess the influence of natural selection on these gene fragments within the South African populations. The ACR locus was excluded from the selection analyses within vervets due to lack of variation in both exons.

To estimate the presence of positive and/or negative selection, two codon-based methods as implemented in the Datamonkey web server were used (Kosakovsky Pond & Frost, 2005). The methods included the MEME method for identifying sites under episodic diversifying or positive selection via a maximum likelihood approach (Murrell et al., 2012), and the Bayesian FUBAR model described by Murrell et al. (2013) which is a superior model to the related likelihood-based FEL model. Rather than reporting evidence of positive selection using p-values, FUBAR uses posterior probabilities due to the Bayesian algorithm used. FUBAR is also capable of analysing large alignments at a high speed, and might outperform FEL if positive selection is present but weak (Murrell et al., 2013; HyPhy, 2018). Datamonkey ignores identical sequences and therefore only calculate selection from the phased haplotype data. Sites were accepted as candidates for selection with p-values < 0.1 for MEME and posterior probability (Post Pr) > 0.9 for FUBAR. The optimal model of nucleotide substitution was selected for each dataset using the Akaike information criterion (Akaike, 1974) in jModelTest (Darriba et al., 2012), prior to the site-by-site selection analyses in Datamonkey. Default settings were used for all other parameters.

An additional selection analysis was conducted in Codeml (Kohlhase, 2006), as implemented in PamlX v1.3.1 (Yang, 2007; Xu & Yang, 2013), using the model pair M7 and M8 to assess site-by-site selection for the three primate sequence alignments. The codon is considered the unit of evolution (Goldman & Yang, 1994) in these models and use the non-synonymous/synonymous rate ration (ω = dN/dS) for selection analysis. It was shown that the likelihood-ration tests (LRT) for this model pair identified sites of positive selection more readily than the alternative M1 and M2 pair. This was attributed to the unrealistic nature of the strict neutral model (M1) which does not account for sites with 0 < ω < 1 (Yang et al., 2000). The F3×4 model of codon frequencies were used in all analyses. A NJ tree obtained from Datamonkey for each gene was used as a working topology. These models were compared for each gene region using a LRT calculated in the program Impact_S (Maldonado et al., 2014). A Bayes empirical Bayes approach (Yang, Wong & Nielsen, 2005) was used to identify codons under selection for model M8. Sites with a posterior probability of > 90% were considered as candidates for selection, as identifying specific sites under selection is more difficult than to identify a specific proportion of sites (Wlasiuk & Nachman, 2010).

Assessment of natural selection within primates

A second set of analyses focused on the ACR, TLR4 and TLR7 sequence alignments of 23 primate taxa and the South African vervet haplotypes from this study. Primate sequences for ACR and both TLR coding regions were downloaded from GenBank. Sequences from 21 primate species were downloaded for TLR4 and TLR7. Only 11 GenBank sequences covered our amplified ACR coding region. The exonic regions for the ACR sequences were identified via the Chlorocebus sabaeus (NC_023660) ACR gene region viewed through the graphical viewer tool from GenBank. See Table S2 for details on sequence accession numbers. The ACR, TLR4 and TLR7 vervet monkey haplotype were added to the GenBank sequences. Sequence alignments were performed using the online version of MAFFT (Katoh et al., 2002; Katoh, Rozewicki & Yamada, 2017), which is more accurate when analysing datasets containing insertions or deletions (Golubchik et al., 2007).

The same Datamonkey and Codeml selection models used in the vervet monkey analyses were implemented to identify sites under selection in the primate dataset. Identical assessment methods were also implemented when considering candidate sites.

Results

Genetic diversity within vervet monkey populations

More than 90% of all vervet monkey samples successfully amplified at the three gene regions. Following trimming of the aligned sequences, a total of 585 bp of TLR4 (partial exon 3), 1,296 bp of TLR7 (partial exon 3) and 1 034 bp of ACR (partial exon 4, intron 4 and partial exon 5) were available for downstream analyses. All haplotypes were deposited on GenBank (Accession numbers: ACR, MG014710–MG014711; TLR4, MG014712–MG014714; TLR7, MG014715–MG014719). A moderate amount of genetic diversity was observed for the two TLR exonic gene regions, with only one polymorphic site observed within the intronic region of the vervet ACR gene region (SNP frequencies: T/T = 0.76; T/G = 0.2; G/G = 0.04). No polymorphisms were observed in either amplified ACR exonic region. For the TLR gene regions we observed two polymorphic sites for TLR4, with one non-synonymous position identified in the vervet population from the C2 region, and three polymorphic sites for TLR7, with one non-synonymous position observed in vervet populations from the N1, NC1 and NC2&3 regions. Contrasting patterns of Hd was observed for the two TLR genes (Table 3). The highest TLR4 haplotype diversities were estimated for the northern (N1, Hd = 0.337) and central (C1, Hd = 0.294) regions, with the highest TLR7 values observed for the northern and northern coastal belt regions (N1, Hd = 0.509; NC1, Hd = 0.425; NC2&3, Hd = 0.660; Fig. 2). Similar patterns were observed for the nucleotide diversity estimates. ACR Hd estimates followed a similar pattern to TLR4 diversity, with the highest levels seen in the northern and central regions (N1, Hd = 0.519; C2, Hd = 0.337) and the dominant haplotype occurring at a frequency of 0.86. One dominant haplotype was observed for both TLR4 (92.47%) and TLR7 (80.77%) across all populations (Figs. 1B and 1C; Fig. 3). The partial ACR, TLR4 and TLR7 sequences generated for this study were modelled onto protein models to view the position of these segments and the identified non-synonymous mutations in TLR4 and TLR7 (Fig. 4). Both non-synonymous mutations resulted in amino acid changes in loops on the protein structure close to ligand recognition sites (Figs. 4B and 4C).

Table 3 Genetic diversity estimates for TLR4 and TLR7 at the eight sampling regions for South African vervet monkeys.

Gene	Population	N	Size (bp)	Polymorphic sites (S)	Number of haplotypes (h)	Haplotype diversity (Hd)	Nucleotide diversity (π)	
TLR4	All primates	24	585	245	23	0.996	0.0838	
All vervet	73	585	2	3	0.141	0.0002	
N1	10		1	2	0.337	0.0006	
NE1	9		1	2	0.111	0.0002	
C1	9		1	2	0.294	0.0005	
C2	10		2	3	0.195	0.0003	
NW1	9		0	1	0.000	0.0000	
SC1	10		0	1	0.000	0.0000	
NC1	8		0	1	0.000	0.0000	
NC2&3	8		1	2	0.125	0.0002	
TLR7	All primates	26	1,299	361	23	0.988	0.0496	
All vervet	80	1,296	3	4	0.330	0.0004	
N1	11		3	4	0.509	0.0007	
NE1	9		1	2	0.282	0.0002	
C1	10		1	2	0.118	0.0001	
C2	10		0	1	0.000	0.0000	
NW1	10		0	1	0.000	0.0000	
SC1	10		0	1	0.000	0.0000	
NC1	10		2	2	0.425	0.0007	
NC2&3	10		3	3	0.660	0.0010	
ACR	All primates$	12	615	176	11	0.985	0.1075	
All vervet*	75	1,034	1	2	0.242	0.0002	
N1*	11		1	2	0.519	0.0005	
NE1*	9		0	1	0.000	0.0000	
C1*	9		1	2	0.294	0.0003	
C2*	10		1	2	0.337	0.0003	
NW1*	8		0	1	0.000	0.0000	
SC1*	9		1	2	0.209	0.0002	
NC1*	10		0	1	0.000	0.0000	
NC2&3*	9		1	2	0.209	0.0002	
Notes:

N, Number of samples; bp, sequence size in base pairs, S, number of polymorphic sites; h, number of haplotypes, Hd, haplotype diversity; π, nucleotide diversity estimates are provided.

* Diversity estimates were calculated from the complete amplified fragment. The single polymorphic site for the ACR region was observed in the intronic segment of the sequence.

$ Diversity estimates only determined for the partial exon 4 and exon 5 regions.

Figure 2 Haplotype diversity estimates for the different vervet monkey sampling localities as calculated in DnaSP.

The black bars represent values calculated for TLR4, red bars TLR7 and blue bars ACR haplotype diversity values.

Figure 3 The minimum spanning haplotype networks estimated for the three gene regions sequenced in the current study.

(A) Acrosin (ACR) haplotype network calculated from the intronic and two exonic regions amplified, (B) Toll-like receptor (TLR) 4 haplotype network and (C) TLR7 haplotype network. The size of each circle is in relation to the number of sequences analysed.

Figure 4 Protein structures of the three genes assessed for genetic variation in South African vervet monkeys.

(A) The partial ACR segment (blue) investigated in this study covered 32% of the reference β-acrosin protein (yellow). (B) The partial vervet TLR4 segment (blue) covered 31% of the Chlorocebus sabaeus TLR4 reference protein B-chain. The known ligand structures are shown as: NAG, purple; MYR, turquoise; DAO, orange; KDO, green. (C) The partial vervet TLR7 segment (blue) covered 52% of the Chlorocebus sabaeus TLR7 reference protein A-chain. The identified U-ligand structure is shown in green. The non-synonymous SNPs are identified by the red circle in B and C.

No significant correlation was found between Hd and geographic distance between vervet monkey sampling localities, following the Mantel analyses (ACR, r = −0.006, p = >0.05; TLR4, r = 0.197, p > 0.05; TLR7, r = 0.208, p > 0.05). The GLM analysis identified a negative, although non-significant, correlation between TLR4 Hd and MAP (Slope = −0.0002; p > 0.05; Fig. 5) and TLR4 Hd and MAT (Slope = −0.004; p > 0.5), with a positive correlation to MaxT (Slope = 0.014; p > 0.05). For TLR7 a positive correlation was observed between Hd and MAP (Slope = 0.001; p < 0.05; Fig. 5). A positive and a negative, although non-significant, correlation was observed for MAT and MaxT respectively (MAT; Slope = 0.053; p > 0.05; MaxT; Slope = −0.031; p > 0.05). The same trends were observed with the GLM analyses based on the non-synonymous TLR Hd estimates. No significant correlations were observed between the neutral D-Loop region and MAP (Slope = −317.24, p > 0.05) and MAT (Slope = 1.867, p > 0.05). A significantly positive correlation was, however, identified between D-loop Hd and MaxT (Slope = 0.093, p < 0.05). See Table S3 for full GLM statistics.

Figure 5 The generalised linear model (GLM) results for the correlation analysis between vervet TLR4 and TLR7 haplotype diversity and mean annual precipitation.

(A) Mean annual precipitation (mm) vs TLR4 haplotype diversity (Hd) and (B) mean annual precipitation (mm) vs TLR7 Hd. N1, Northern; NE1, North eastern; NW1, North western; C1, Central 1; C2, Central 2; NC1, Northern coastal belt 1; NC2, Northern coastal belt 2; NC3, Northern coastal belt 3; SC1, Southern coastal belt 1.

Role of selection within vervet monkeys

For the vervet TLR4 alignment, only one codon was identified by the FUBAR method as possibly being under negative (purifying) selection, which is a non-synonymous mutation at codon 144 (human codon 359; nucleotide 432, T -> C). A strong indication of negative selection at one TLR7 codon (codon 419, human codon 500, nucleotide 432, C -> T) was identified with FUBAR (Table S4). This polymorphism is, however, a synonymous mutation and the non-synonymous mutation at TLR7 codon 402 (human codon 483) was not identified as a candidate for selection. These codon positions are based on our amplified fragments lengths. No codons under positive selection were identified.

Genetic diversity within primates

All three primate sequence alignments showed high levels of genetic diversity, with 176, 245 and 361 polymorphic sites observed for ACR, TLR4 and TLR7 respectively (Table 3). The highest number of polymorphic sites were observed for TLR7, and the highest Hd levels were observed for TLR4 (Hd = 0.996; Table 3).

Role of selection within primates

Within primates, eight codons showed signs of positive, and possibly episodic, selection for the ACR coding region following the MEME results. FUBAR only identified one candidate site for positive selection, with M8 identifying two candidate sites. Furthermore, 21 ACR codons were identified as potentially under negative selection. Two and seven sites of potential positive selection were identified for TLR4 using FUBAR and M8 respectively. Twelve candidate sites for episodic positive selection was identified by the MEME analysis. Five sites of possible negative selection were identified for TLR4 following the FUBAR analysis (Table S4). Between the two immune-linked genes, TLR7 displayed the highest amount of candidate sites for positive selection (n = 6) identified with FUBAR, but only three sites were identified with M8. Ten sites possibly under episodic positive selection was identified with MEME, with 52 candidate sites under possible negative selection.

Discussion

The genetic diversity estimates for South African vervet monkey immune-linked genes showed significant correlations to environmental factors associated with specific bioregions within South Africa. Host–pathogen interactions can shape the genetic variation observed in host immune-linked genes (Hamilton, 1982; Little, 2002; Zhan et al., 2002; Villaseñor-Cardoso & Ortega, 2011). Pathogen prevalence and abundance has also been linked to climatic factors such as precipitation and temperature (Guernier, Hochberg & Guégan, 2004; Harvell et al., 2009; Lafferty, 2009; Dunn et al., 2010).

Vervet monkey variation and selection

The higher TLR7 diversity observed along the Indian Ocean coastal belt had a strong positive correlation to the high mean rainfall observed in this area. The only non-synonymous mutation observed in this gene region was also seen in the Indian Ocean coastal belt populations. A number of studies have linked higher rainfall to increased human and non-human primate pathogen prevalence (Kupek, de Sousa & de Souza, 2000; Guernier, Hochberg & Guégan, 2004; Chapman et al., 2010; Gaetano et al., 2014). A study on African green monkey SIVagm infections in South African vervet monkeys, showed a distinct geographic structuring of SIVagm genetic strains across South Africa (Ma et al., 2013). Three main clusters were identified; one with only Free State (Grassland biome) strains, one with Eastern Cape (Albany thicket biome) and KwaZulu-Natal (Indian Ocean coastal belt) strains and a third with mostly KwaZulu-Natal strains. According to the Ma et al. (2013) study one can infer that the vervet monkeys from the Indian Ocean coastal belt region are exposed to a wider variety of SIVagm strains. Villaseñor-Cardoso & Ortega (2011) suggested that a higher level of variation at immune-linked loci in a population could lead to an increased potential for a population to respond to a wider range of pathogen-associated molecules. A higher level of pathogen diversity could therefore support the occurrence of a higher TLR7 gene diversity in this area. It was also observed that the non-synonymous mutation identified in our TLR7 dataset occurs close to a uridine-5′-monophosphate ligand recognition site (Fig. 4C). This site is crucial for TLR7 recognition of viral ssRNAs (Diebold et al., 2006; Zhang et al., 2016) originating from viruses such as SIV (Kaup et al., 2005), HIV-1 (Heil et al., 2004; Simmons et al., 2013), vesicular stomatitis virus (Lund et al., 2004), hepatitis C virus (Lee et al., 2006) and influenza virus (Lund et al., 2004; Wang et al., 2008).

The negative correlation between vervet monkey TLR4 regional genetic variation and the MAP, although insignificant, was interesting. Higher rainfall is generally linked to higher pathogen diversity (Lafferty, 2009), which in turn could lead to higher TLR diversity. Higher TLR4 Hd values were, however, observed in the central regions than seen in the Indian Ocean coastal belt regions and the only non-synonymous mutation was also observed in the central region. A number of possible ligands were observed close to the non-synonymous mutation site identified for the vervet TLR4 sequences (Fig. 4B). All of these ligands are associated with gram negative bacteria. The monosaccharide N-acetyl-d-glucosamine (NAG or GalNAc) forms part of the O-specific polysaccharide of gram-negative bacterial lipopolysaccharides (LPS) (Kilár, Dörnyei & Kocsis, 2013). NAG is also an essential component of fungal cell walls (Lee et al., 2008) as well as some parasites (Araujo, Souto-Padrón & Souza, 1993) and is excreted by mucolytic bacteria (Sicard et al., 2017). The ulosonic acid 3-deoxy-d-manno-oct-2-ulosonic acid (KDO) is a component of the LPS inner core of most gram negative bacteria (Kilár, Dörnyei & Kocsis, 2013), as well as the inner core of LPS of enteric gram-negative bacilli like N. meningitidis (Tzeng et al., 2002). The saturated fatty-acids lauric acid (DAO) and Myristic acid (MYR) forms part of the Lipid A component of LPS (Lee et al., 2001; Bäckhed et al., 2003). Lipid A is known as an endotoxin responsible for the pathogenicity of gram negative bacteria (Raetz et al., 1991; Ribeiro, Zhou & Raetz, 1999; Tzeng et al., 2002).

Bacterial abundance and diversity will vary among the different regions, with some species more prevalent in drier areas than others. Patz et al. (1996) observed that meningococcal meningitis epidemics, in sub-Saharan African human populations, predominantly occur during the hot dry season. This disease is mainly caused by the gram-negative bacteria N. meningitidis (Brandtzaeg, 2006). The highest MaxTs were also notably observed for the central regions (Table 1) and could affect TLR4 diversity in vervet monkeys from this region. Numerous heat shock proteins are associated with TLR4 activity and vice versa (Ohashi et al., 2000; Asea, 2003; Roelofs et al., 2006). Small Heat Shock Protein B8 (HSP22) have been identified as a ligand for TLR4 during inflammation caused by rheumatoid arthritis (Roelofs et al., 2006) and HSP70 is known to have chaperokine activity by interacting with TLR4 following cellular stress such as pathogen exposure or exercise (Asea, 2003). Ohashi et al. (2000) suggested that TLR4’s murine ortholog, Tlr4 is a mediator for heat shock protein 60 (HSP60) in mice. Further analysis of different HSP genes could provide more insights to the relationship between heat shock proteins and TLR4 diversity and regional temperature differences across the vervet monkey distribution range. A combination of TLR4 mediation of heat shock proteins and exposure to gram-negative bacteria more prevalent in drier areas could in theory drive the retention of different haplotypes in these areas. Divergence of vervet monkeys from the central region from those in the coastal regions was estimated at 1.099 million years ago (Turner et al., 2016b). Immune genes generally evolve at a faster rate than other genes (Hurst & Smith, 1999), and therefore the time since divergence would have been sufficient to support the occurrence of the observed haplotype variation between the two regions. The different patterns of genetic diversity observed between TLR4 and TLR7 highlights the intricate role different environmental factors or pathogen types (bacterial vs viral) can play on immune gene evolution. More in-depth sampling and larger sequence lengths, as well as a study on bacterial diversity in vervet monkey populations, might provide a better understanding of TLR4 diversity in these primates and the role these factors might play.

The positive correlation observed between the mitochondrial D-loop and MaxT could be due to the mitochondrial involvement in cell metabolism (Wilson et al., 1985). Lovegrove (2003) showed that ambient temperature has a strong effect on basal metabolic rate. The correlation of mtDNA D-loop variation and MaxT could thus be due to the role this region plays in replication and expression of mitochondrial genes (Sbisà et al., 1997), such as cytochrome b which is involved in metabolism and forms an important component of the mitochondrial electron-transfer chain (Zhang et al., 1998). This result will need to be assessed in further detail, either through sequencing of targeted metabolic genes or through whole mitochondrial genome sequencing of specimens from the study localities. These additional sequencing data will make it possible to better understand the link between climate and mtDNA variation.

Failure to identify signs of positive selection at the two vervet TLR gene segments can be attributed to the low number of haplotypes (TLR4 haplotype n = 3; TLR7 haplotype n = 5) identified for analysis. The lack of positive selection at TLR genes within species could also be linked to the occurrence of episodic selection (Wlasiuk & Nachman, 2010). Episodic positive selection is a known consequence of pathogen-mediated selection as pathogen infections might be more sporadic in nature than long established events (Wlasiuk & Nachman, 2010; Grueber, Wallis & Jamieson, 2014). Sites under episodic positive selection will mainly be under strong purifying selection during their evolution, with bursts of strong positive selection occurring in some lineages (Murrell et al., 2012). These sites can, however, be masked by the overpowering signal of purifying selection that is at play. The MEME analysis for the detection of episodic selection, however, did not identify any sites under episodic selection. Small sample size could, however, also have affected this analysis.

A number of studies have suggested that diversity at synonymous positions could in fact have adaptive significance and influence protein expression (Chamary, Parmley & Hurst, 2006; Hunt et al., 2014) by either changing splice sites (Pagani, Raponi & Baralle, 2005; Parmley, Chamary & Hurst, 2006; Batista et al., 2017) or mRNA secondary structure (Duan et al., 2003; Chamary & Hurst, 2005; Chen et al., 2017). The observation of possible purifying selection at a synonymous position at TLR7 further supports this view. We were, however, unable to identify the possible effect these synonymous mutations have on TLR7 expression. This will have to be assessed in further analyses.

The ACR gene show significant levels of variation between mammalian species, with exon 5 being the most rapidly evolving segment (Gatesy & Swanson, 2007). The mammalian ACR gene is, however, functionally conserved across mammalian species, highlighting its role in mammalian fertility (Raterman & Springer, 2008). Lack of variation within the amplified vervet ACR fragment and the disproportional occurrence of one gene variant, could be linked to this conservation of functionality within the vervet monkey species. Natural selection would favour sperm which contains a functionally stronger gene variant, which will outperform other variants leading to a selective sweep, resulting in a dominant variant remaining in the species. Metz, Robles-Sikisaka & Vacquier (1998) suggested that the occurrence of a selective sweep might have led to the observed lack of genetic diversity at the reproductive gene lysin in red abalone species (Haliotis rufescens). Further analysis of gene regions located close to the vervet ACR gene will be required to verify the occurrence of a selective sweep at this gene, as genetic hitchhiking is a known consequence of selective sweeps (Choudhuri, 2014). An assessment of the level of ACR gene differentiation between and within all Chlorocebus species would also be an interesting future study. We only had one ACR sequence from Chlorocebus sabaeus, which showed a 21 bp indel in the proline rich region of exon 5 when compared to our vervet ACR sequences. Additional analyses could shed more light on the evolution of this gene within the genus Chlorocebus.

Variation and selection within primates

The high number of sites under negative selection at TLR7 supports the observation by Nakajima et al. (2008) that TLR7 is subject to negative/purifying selection in Old World primates. Within primates, TLR7 had, overall, the lowest amount of positively selected sites compared to TLR4 which is also congruent with previous findings Wlasiuk & Nachman (2010). The small difference in number of sites under positive selection between TLR4 and TLR7 in our study can also be attributed to the smaller sequence length of the TLR4 region in our study compared to TLR7. The work by Wlasiuk & Nachman (2010) focused on the complete coding regions of primate TLR genes and found a larger difference in the number of sites under selection for these two genes. Fornůsková et al. (2013) observed a similar trend between rodent Tlr4 (Tlr4 polymorphic sites = 545) and Tlr7 (Tlr7 polymorphic sites = 466) sequences while assessing the genetic variation and evolutionary processes involved in these two genes and 23 Murinae species. A study assessing the autosomal sequence variation of European and African Bos taurus and Asian Bos indicus indicus revealed that TLR7 showed no genetic variation in any of the breeds, whereas TLR4 showed higher than average signs of genetic variation in Bos indicus (Murray et al., 2010). The higher level of positive selection observed for TLR4 vs TLR7 can be associated with stronger selective pressures playing a role on non-viral TLR loci (TLR4) compared to viral TLRs (TLR7) in primates (Nakajima et al., 2008; Wlasiuk & Nachman, 2010).

Between species variation for ACR was prominent, with numerous indels observed within the proline rich region of exon 5. This proline rich region is cleaved during the conversion of the zymogen form proacrosin to the mature form (Adham, Schlösser & Engel, 2004). A comparison between various artiodactyl ACR exon 5 sequences by Gatesy & Swanson (2007), however, showed no signs of indels within this region. This could be explained by different evolutionary and life histories of artiodactyls and primates. Different reproductive strategies could be a driver for this difference, with artiodactyls producing small numbers of large, well-developed, offspring capable of escaping predators and primates giving birth to a few large offspring which needs to be carried for protection. This behaviour limits the number of offspring primates can produce per season (Sibly & Brown, 2009). Sibly & Brown (2007) also showed that artiodactyls have higher reproductive rates than primates. This was linked to primates occupying areas with lower predation, which is linked to the number of offspring needed for population growth (Sibly & Brown, 2007). Raterman & Springer (2008) observed no sites under significant positive selection at ACR for placental mammals. In contrast to the results from Raterman & Springer (2008), signs of positive selection for ACR in mammals were identified by Swanson, Nielsen & Yang (2003) supporting the nine sites of possible positive selection identified within our primate ACR dataset.

Signs of strong negative selection was observed in all three genes. Negative or purifying selection is responsible for the removal of deleterious mutations from the genome, and thereby preventing the accumulation of these mutations (Fay, Wyckoff & Wu, 2001). This can be linked to the importance of maintaining stable functional copies of immune-linked and reproductive genes in a population. These strong signals of negative selection can also potentially mask the occurrence of sites under episodic positive selection (Murrell et al., 2012).

Conclusion

Results from the current study confirm that adaptation to some environmental factors (rainfall and pathogen prevalence) can be linked to genetic variation observed at different immune-linked genes. Specifically, vervet TLR diversity could be shaped by environmental drivers linked to pathogen abundance and prevalence. Vervet monkey TLR4 gene diversity might be shaped by gram-negative bacteria linked to drier climates, as well as its involvement in heat shock protein activity. Higher rainfall and possible increased pathogen prevalence in high rainfall areas, were linked to the observed TLR7 gene variation for South African vervet monkeys. These contrasting patterns of gene diversity coincides with the strong genetic structuring previously observed in vervet monkey mtDNA diversity. All three genes (ACR, TLR4 and TLR7) are under strong selective pressures within primates. The clear differences in evolutionary patterns observed between the two TLR genes might best be explained by the location of expression, as well as the different types of pathogens they respond to. The observations from this study add valuable information to the already existing knowledge surrounding South African vervet monkey evolution and could be valuable for future research in biogeography or host-pathogen ecology. These results will also help to assist conservation agencies to better plan re-introduction or translocation programs for rehabilitated animals, if needed. The genetic structuring linked to selective pressures associated to different environments should therefore be considered when identifying appropriate re-introduction sites or recipient populations.

Supplemental Information

Supplemental Information 1 Haplotype DNA sequences for vervet ACR, TLR4 and TLR7 generated during the current study.

Click here for additional data file.

Supplemental Information 2 Table S1. Sample list of 81 vervet monkeys sampled from across the vervet monkey distribution range for the current study.

Locality information, group ID, no of troops per group, sex, sample ID and DNA amplification success for each gene fragment is provided.

Click here for additional data file.

Supplemental Information 3 Table S2. Sample list of all outgroup taxa used during analyses, with the associated GenBank accession numbers.

Click here for additional data file.

Supplemental Information 4 Table S3. The statistical results from the Generalized linear model (GLM) analyses performed on the vervet monkey ACR, TLR4, TLR7 sequences from the current study and D-loop DNA sequences from Turner et al (2016b).

Values of very small magnitude were rounded to four decimal places.

Click here for additional data file.

Supplemental Information 5 Table S4. Site-by-site results for selection obtained from three different selection analyses.

Significance values (p-value/Posterior probability) are in bold. Candidate sites identified are highlighted in grey.

Click here for additional data file.

The authors would like to thank all those involved in the original sample collection for the Turner et al. (2016b) study. Specifically, Dr Joseph G. Lorenz and Dr Nelson B. Freimer for their contributions towards the larger vervet monkey project. We would also like to thank Elmarie Blom and Rudi Lombaard for their assistance in the laboratory.

Additional Information and Declarations

Competing Interests

Author Contributions

Animal Ethics

DNA Deposition

Data Availability

The authors declare that they have no competing interests.

Willem G. Coetzer conceived and designed the experiments, performed the experiments, analysed the data, contributed reagents/materials/analysis tools, prepared figures and/or tables, authored or reviewed drafts of the paper, approved the final draft.

Trudy R. Turner contributed reagents/materials/analysis tools, approved the final draft.

Christopher A. Schmitt approved the final draft, significant contribution to sample and data collection.

J. Paul Grobler conceived and designed the experiments, contributed reagents/materials/analysis tools, authored or reviewed drafts of the paper, approved the final draft.

The following information was supplied relating to ethical approvals (i.e. approving body and any reference numbers):

Sampling and research ethical clearance was obtained for the original paper (Turner et al., 2016b) from the Institutional Care and Use Committee of the University of Wisconsin–Milwaukee (Ref no. 07-08 #32) and the Inter-Faculty Animal Ethics Committee of the University of the Free State, South Africa (Ref no. UFS-AED13/2010). All samples from this study were sourced from the Turner et al. (2016b) study and no new additional samples were included.

The following information was supplied regarding the deposition of DNA sequences:

All sequences generated are accessible via GenBank, accession numbers MG014710–MG014719 and in the Supplemental Information.

The following information was supplied regarding data availability:

The raw data are provided in the Supplemental Files.

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
