# Peer review of "Adaptive genetic variation at three loci in South African vervet monkeys (Chlorocebus pygerythrus) and the role of selection within primates"

_PeerJ, doi:10.7717/peerj.4953_

## Round 0.1 · original submission · Major Revisions

All three reviewers agree that the manuscript was well-prepared and constitutes a strong contribution to the scientific literature. The sample size is quite large for wild sampling and the attempt to link specific environmental conditions to possible selection events in a range of vervets is interesting. One reviewer notes that the introduction could be revised to include greater context on adaptation and variation within the genus more broadly. I agree that the introduction requires substantively greater context than provided including the specific contributions as to how this will aid conservation efforts. The main problems that lead to a major revision decision arise from methodology which may not constitute a serious flaw but will require substantive revision to address key concerns surrounding data generation, analytical approaches, and confounding factors. These problems may stem from an over-reliance on the Turner 2016 paper as a citation (from which this analysis is extracted) rather than explaining afresh the detailed methods for data generation and analysis including choices made. The paper generally relies on a citation, leaving the reader to follow up independently on highly relevant information (such as the target sequences and environmental data as two of many examples). A revision must strike a balance between not replicating published efforts but summarizing the important and relevant information from those sources to satisfy readers that the research pathway is sound and can be replicated using information contained with this particular paper. And, a revision needs to clearly link the larger picture of what we know to the goals and outcomes of this study with greater detail on choices and methods and stronger links between the immune adaptive focus and the environmental genetic variation focus.

Data generation is not clear. The source of DNA and extraction methods should be more clearly explained (possibly a summary table of individual and type of tissue). More importantly, the targets are not clearly referenced, defined, or described. This is particularly important because the targets are sometimes only partial rather than full gene. Given that so few areas area examined, the partial sequences are particularly troubling without insights into why those areas were chosen.
Analytical approaches need greater justification and clarification. Two reviewers note that data are limited and potentially over-analysed and results over-interpreted—inferring larger conclusions such as the selective sweep due to the observation of low diversity. The authors must provide greater context and explanation of their interpretations to convince the reader of their plausibility. The authors should clearly explain the choices made and why important genes noted in the literature review were not considered in their research design. I see that the introduction winnows down the larger set of genes that could be examined to two that genes that do not cause AIDs symptoms in hosts but the leap to lack of AIDs symptoms as inclusion criteria is too great to follow. Very little was discussed about the mtDNA which may be why one reviewer questioned the D-loop discussion. Ultimately, the tests are presented as a series of data manipulations rather than a well-designed study seeking to answer specific questions as assessed by specific tests.

A few confounding factors noted are the X-chromosome genes and possible relatedness of animals sampled, both of which require additional measures for analysis. The X-chromosome TLR7 is discussed in the text as limited to haplotype analysis for females given the lack of two copies in males but then data are lost for males taking this approach.

Replicability is increasingly an issue that has arisen from the modern wave of open access data sharing. The reviewers note some problems that would hinder replicability of this study and more reporting of raw data is requested. Some samples were not found in GenBank and should be added prior to publication. Please pay attention to the detailed comments in each review as they point to missing citations, incorrect characterization of key research elements, formatting, and confusing language.

Reviewer 1 ·

Basic reporting

Basic reporting is acceptable.

Experimental design

Experimental design is appropriate, although analyses would benefit from a more targeted approach - see my general comments.

Validity of the findings

The findings and interpretation are largely in line with the results. See also my general comments.

Additional comments

In this study, the authors examine diversity of three genes (TLR4, TLR7 and ACR) in vervet monkeys across South Africa. Population genetic analysis was undertaken to examine the distribution of diversity across habitats, although overall diversity was low. The researchers also examined sequence variation among species by comparing their new data to sequences from 11 – 21 primates and conducting various selection analyses. Overall this paper is well written, however the analyses are hampered by very low levels of diversity overall (although I note that low diversity should not necessarily preclude publication). The addition of cross-species comparisons provides an additional angle to the work that supports the main findings. I have a few comments that relate to the analysis:

At present, the paper comes across as though the authors attempted to conduct as many analyses as possible, to try and make the most of the data they had. While it is commendable that the authors should explore all aspects of their data, unfortunately when diversity is low this also poses a risk of over interpreting data. I believe the paper would benefit from a more targeted approach to the analysis. For example, I am not sure that population-level substitution analyses (such as Tajima’s D) would be informative for a dataset that contains only a handful of SNPs, although welcome the authors’ perspective on this. Similarly, seven codon-based sequence analyses are conducted (5 datamonkey methods, plus 2 Paml methods) – this is a lot. Many of these methods overlap in their objectives, and are therefore not necessarily independent lines of evidence for selection. It would be helpful to point out which methods are similar to one another, and justify why using each is important. What different information do they provide? What are the pros and cons of each method?

The authors allude to a “selective sweep” as a possible explanation for the low diversity they observe (mentioned in the Abstract, and otherwise just once, in the Discussion at L355). This is an interesting idea relating to the low diversity that was observed, and which could therefore be expanded. What evolutionary mechanisms would cause a sweep for these genes? Is it likely in this widespread species? Are there precedents in the literature? What additional data would be required in order to confirm the sweep?

Do the statistical methods used throughout the paper account for the fact that TLR7 is on the X-chromosome (and that males therefore are hemizygous, not homozygous)? This is important, because much of the ecological analysis relies on haplotype diversity, which is calculated from haplotype frequencies and therefore depends on an accurate count of gene copies in the population. It would be appropriate to reiterate the suitability of methods for sex-linked loci throughout the Methods.

Minor comments:

L26: “Low levels” – be explicit, give the numbers of synonymous and nonsynonymous SNPs
L35-36: Briefly explain how this data could be used.
L109: Suggest changing “the original paper” to “a previous study using mtDNA analysis”
L222: In the current version, the source of the samples is explained 3 times – suggest deleting from here to avoid repetition
L129: Explain how these “partial segments” were chosen for targeting
L172: Explain what kind of GLM was used and the rationale – was it a logistic regression?
L290: “It is well known…” please provide references and context
L312: Why is the non-significant result surprising? The paragraph L311-332 draws a lot of inference from a non-significant finding. I believe it would be more appropriate to shorten this substantially.
L334: I do not understand what information is being conveyed in the parentheses.
L343-347: Note that purifying selection is identified at the codon level, not the SNP position. Thus, I’m unclear how these results support evidence that synonymous substitutions could have adaptive significance. It is the codon that is under purifying selection.
L380: It would be useful to explain how the different evolutionary histories of these taxa could drive the different patterns – what evolutionary process would you hypothesise?
L390: Typo “selection”
L394: Missing word “different immune-linked genes”
L400-401: As “conservation” is also a rationale for this study (Intro L106), it would be helpful to have some information on the conservation status of the species and current threats. Please provide more detail of how the results of the current study could be used to inform conservation of this species.
Figure 1: Please explain the importance of the difference symbols used for the sampling sites
Figure 1: The position of the sampling points in C is shifted relative to their position in B – is this intentional?
Figure 3: According to the hypotheses posed, I think it would be more appropriate if Hd were the dependant variable (response, y-axis) and precipitation were the independent variable (predictor, x-axis) – is this how the statistics were conducted?

Reviewer 2 ·

Basic reporting

The manuscript is well-written in clear, professional language. Although the figures and tables clearly present the materials and the outcomes of the analyses, the presentation of additional data would be useful, especially the following.
• A table should be added that summarizes all vervet polymorphic sites that were identified in the study.
• Raw sequence data is unnecessary, but an additional supplementary table containing genotype data from all sequenced animals would be useful.
The Introduction well summarizes the background information and the rationale behind this study; however, it could be supplemented with what is known about the genetic diversity and adaptive genetic variation in the genus Chlorocebus.

Experimental design

The study is original research, within the scope of the journal, and approved by the appropriate ethics committee. The overall experimental design is clear and implements standard methodology that properly addresses the research question about the effects of selection on genetic variation in South African vervet populations. The study takes an interesting approach, combining genetic data from vervet monkey populations and local environmental data supplemented with natural selection analyses in vervet populations and 23 other primate taxa. The study targets three preselected candidate genes that are potentially linked to fitness and climatic conditions. A limitation of this study is that it analyzes only partial gene sequences, including only 1‒2 exons with adjacent introns.
• Line 224: Haplotype sequences mentioned here as being deposited in the GenBank are not available in this database. Please add them.
• How were the target sequences within the selected genes chosen? If it is available, provide information on the functionality of the gene region covered by this analysis. Do they code any specific protein domains?
• Supplementary Table 1 suggests that multiple animals were drawn from each of several troops. Comment on the possible effects that the relatedness among the animals in this study has on its results.

Validity of the findings

The data are robust, and the findings are valid. The authors properly comment on the possible limitations and biases that are due to analyzing partial gene sequences and to the relatively low sample size for each haplotype (the overall sample size is impressive for studies in wild primate populations). This study generates important findings beyond vervet monkey evolution. The data obtained can inform primate management in South Africa.

Other Points
• Supplementary Table 1: The “Shamwari NR, Dump, EC” site lacks geographic coordinates.
• Figure 3: It would be useful to include p-val and slope in the figure legend.
• Supplementary Figure 1: Clarify in the figure legend whether it is a haplotype analysis.
• Line 28: The sentence “Population genetic…” seems too vague for the Abstract.
• Line 267‒270: It would be easier to read the text if both the text and Table 3 used the same codon numbering system. For clarity, perhaps you could just add human codon numbers to the text.
• Line 276‒279: Based on Table 3, the highest haplotype diversity levels seem to be for TLR4 (0.996), not TLR7 (0.988), as stated in the text.
• Line 305: To which study does “from this study” refer?

Reviewer 3 ·

Basic reporting

Overall the language is clear and professional, but there are some areas that need clarification or correction, cited here for convenience

1) Lines 35-36 regarding impact of the study on conservation agency decision making are not very clear. Please reword to better connect to the rest of the abstract. The impact described is not obvious.

2) Line 76 the organisms listed here are not PAMPs. PAMPs are pathogen-associated *molecular patterns *. Fix this error by replacing "gram-positive bacteria,..." etc. with the relevant PAMPs (i.e. peptidoglycan etc). It would helpful to put the pathogen type in brackets alongside those PAMPs.

3) Since the paper largely focuses on two TLRs and proposes connections between gene diversity, environmental conditions and the pathogens-detected by these TLRs, the authors should be citing original citations when describing the PAMPs in lines 75-77 rather than a single review. Invivogen has actually done most of this work for you. They state the original papers identifying these molecules as ligands for TLRs in their product inserts for TLRs. That's how you can find those papers, review them and cite them (as you have later in the paper, e.g. Kurt-Jones et al. 2000; Diebold et al. 2004).

4) Related to #2 and #4 lines 77- 79 describing what the two major groups of TLRs recognize is not accurate. Extracellular TLRs recognize a broad range of extracellular threats, including bacteria, parasites, fungi, not "mainly bacteria". TLR4 also recognizes respiratory syncytial virus. Similarly, intracellular TLRs recognize mainly nucleic acids, including self-nucleic acids. So while they certainly recognize viruses, they also recognize unmethylated DNA from intracellular bacteria, fungi, parasites and have been found in experimental to recognize host DNA. Revise this description as "extracellular threats" with examples and "intracellular threats" with examples, and the misleading writing here will be resolved

5) lines 83-86 The broad statement that Gram-negative bacteria can cause a range of infections is poorly backed up by examples and citation. Given the focus on TLRs as receptors, these examples should be name pathogens with the manifestion following. As worded it sounds like E. coli is major cause of meningitis.

6) Throughout the article when genes are clearly being discussed, they are not italicized in text as is required. Please fix this

7) Line 253 possible mistyping. "D- loop". TLR7 has a Z-loop. Is that what you mean here? It otherwise sounds like you are discussing mtDNA, which has not yet been brought up in the paper.

8) descriptions of sources of DNA, here described only as "tissues" and unattributed "samples" are not specific enough to help a reader understand the amplification methods you then pursued. Please describe these samples further in text, or provide a supplementary table. What are the tissues? Are any of these samples not tissues but
fecal?

9) Line 302-310 This section requires develop. The connections being made between environment and "pathogen-specific" molecules are not obvious. Moreover, the paper focuses on TLRs, which detect "pathogen-associated" not "pathogen-specific" molecules, so the meaning of this section is a little confusing given prior pages. This distinction matters because TLRs are innate immune broadly sensing sentries of threats, and the term "pathogen-specific" is usually reserved for descriptions of adaptive cellular responses to an identified epitope.

10) Line 317. TLR4 also detects a range of heat shock proteins in humans (e.g. hs70), which is likely a more relevant example. In both mice and humans TLR4-mediated responses to heat shock proteins are very well established and very common.

11) and probably most importantly, there is a complete lack of clarity regarding the regions of these genes that being amplified. Given that the paper is focused on diversification and evolution of a very small number genes based on only partial amplification of those genes, the bp positions within the gene and the genome sequence should be identified here. Simply stating partial sequence from exon 4 and not detailing in a table or anywhere in the paper where that sequence is in the gene makes it very difficult for a reader to assess the results. Along with that, the domains of the TLRs in this paper are well defined. The domains of these genes have been found to be evolving at different rates. The authors need to clarify in the primer table what domains are being addressed here. Otherwise the results can't be fairly interpreted by the reader.


Article structure is professional, figures, tables are shared. Raw data has been submitted to Genbank

Paper is self-contained with results that inform the hypothesis.

Experimental design

The primary research is within the aims and the scope of the journal. The research question is well defined and relevant an meaningful. The authors have stated that the research fills a knowledge gap for this species.

From an investigative standpoint, though, the paper has problems. Along with the lack of clarity regarding methods, mentioned above, there are some other issues that need to be addressed. The authors analyzed the sequences of only 3 genes from 81 samples. One gene is reproductively linked and the other two are well studied genes associated with the immune system. This is very little data to draw some of the conclusions that the authors have drawn here. If they did test other genes and did not get a result, it would be preferable to list those genes as well. Otherwise, the paper appears insubstantial. Given the relatively low cost, it is difficult to understand, for example, why they did not address other TLRs that are more strongly connected to the background examples they give as a justification for this research. Those papers make connections between climate and parasite infections, for example. Additions like TLR2/1, TLR5, TLR9 would have been logical genes to include.

More importantly no explanation is given for why only partial sequences of genes are examined here. For example, TLR4 is a ~6100 bp gene encoding around 820 amino acids. It's a small readily amplifiable gene, the domains differ significantly in their variability within and between species. The TIR domain and small cytoplasmic tail of TLR4 appear to be able to handle a great deal of variation, with humans, rhesus monkeys and mouse, for example, differing in the length and construct of the this domain considerably. Between rhesus monkeys and humans, for example, a quick alignment of publicly available sequences shows a 13 amino acid deletion in this area. Within species differences can be substantial as well (e.g. this old review. Please note that LRRs are structural repeats, not domains. https://www.ncbi.nlm.nih.gov/pmc/articles/PMC2215765/)

The authors acknowledge that their results may have been shaped by the regions of the gene that they amplified. They most certainly will be, and this has been established in the past, so there needs to be a justification for why only partial gene regions were assessed, what protein domains they represent and why those regions were chosen. This problem holds for TLR7 and ACR as well. Once described, it would be preferable to put the results in the context of what is already known about the evolution of these sequences in mammals or primates, regarding the diversity of the sequences from these domains.

Validity of the findings

See above. The findings are not valid if the authors cannot address why they used partial sequences and put it in the context of what is known about the variability of the domains they amplified either across or within primates, mammals. There could be perfectly reasonable rationale. It needs to be up front.

Data is not particularly robust at the gene level. Only three genes are assessed, when the question would be better addressed by the consideration of more genes. At the within species level, the authors have a very robust data set of 81 samples. They should make clear whether or not those 81 samples represent 81 individuals.

The analysis and are discussion are very simple. Though the results are minimal, there are means of demonstrating their importance to the reader that the authors have not used here. It is strongly suggested that the authors map a reference Chlorocebus sequence to the human TLR4 (3FX1; 4G8A) and TLR7/8 (4QC0) sequences in PDB (there is a boar ACR sequence) and highlight where the variation is occurring in the protein (where applicable). This can be done by homology modeling which is very straightforward. Since the model is not to be assessed or argued, it wouldn't need to be verified. It would serve as a mannequin on which to illustrate a) the data that was actually gathered and b) where the variable hotspots are. Given that the available structures are bound to ligands, this would allow the authors to discuss the implications for pathogen detection. If the amino acid sequences are very similar to humans, then the authors should use a human construct label it as such. Regardless, there is a need to connect the experimental findings to the data that was recovered in a stronger way. This is the method suggested here.

Additional comments

This is an interesting pursuit. To be accepted, the methods of the paper need to be more clear. The rationale for the limited sequence length needs to be stated. The results needs to be interpreted in the context of the protein domains in which the sequence occur because it is known that these domains vary in their diversity between and within species and by domain.

---

## Round 0.2 · Minor Revisions

The reviewers and I agree that the manuscript has met the critical errors identified in the first submission and constitutes a contribution to the literature. Thank for taking the time to revise the manuscript and resubmit!

There are still a few lingering concerns noted by one reviewer regarding the abstract reporting and the paper findings. The manuscript will not be fundamentally altered by addressing the remaining concerns but I cannot recommend it for final publication until the many writing problems are resolved. Ultimately, the writing is hurried, ungrammatical (e.g., verb-subject agreement, run-ons, dangling modifiers), and lacks the clarity required for the complexity of analyses taken to examine these three genes. I have tried to note as many points as I could when making my decision but I am sure I have missed several. I would highly recommend seeking a reader not connected with the material to have a final editorial overview to catch any remaining errors. The goal in addressing these points is to aid the reader in ease of understanding.

Paragraphs at 130 and 140: I am not clear on what contribution these paragraphs are making to the introduction. The content appears to be a review of a limited number of selection tests rather than an explanation of the tests/programs used (which would actually belong in the methods). on 283, only 2 tests are used from Datamonkey which further makes me question the general review here.
172: tense shift--please go through the manuscript and ensure the tense is consistent within each section
266: What is the source of these data? Is this the Line 276 reference? If so, move that reference to where the data are introduced (and perhaps make a comment on whether they are still useful since the publication is 12 years old).
279: this is a one-sentence paragraph. Please fix.
284: verb-subject agreement methods-were, not methods-was
298-299: There is a repeat in the first few words of the sentence. Please fix.
304: This should be 'a second set of analyses'
308-309: how useful it is to take human exons and apply them to monkeys? Just a sentence here justifying or providing some context for this approach.
314: 'The same'--same as what. There are only two sentences in this paragraph with the first one referring to something in previous text. This is one of many dangling modifiers in the paper that need correction.
Dangling modifiers noted in paper (but check for others that I missed):
-‘it is’: 47, 90, 386, 415, 423, 436, 438
-‘it can’: 402, 407
'it was' 417; 'it has' 433; 'it should' 477; 'it has' 479; 'it will be' 497
Please rephrase the above (and any missed) in active voice. Example error from 386: "It is well known that pathogen host interactions can shape..." Possible fix: Pathogen-host interactions can shape OR have been reported to shape OR have capacity to shape OR have been observed to shape.
321: why is the intron included in analysis?
387: affect not effect here. But, ultimately, both halves of this sentence are saying the same thing. I would probably shorten the sentence noting that host-pathogen interactions increase variation in immune system genes (all citations). By increasing variation in one part of the genome, overall variation/diversity is obviously affected. If one set of papers is arguing that other genes increase in diversity (due to hitchhiking or some other process) when the immune system ones increase in diversity, then be clearer about what you mean here. Since you are focused on a few immune genes, I think this clarity is particularly important for the reader to understand what you think the implications of your findings are relative to the broader field.
407: There is a period and space missing here.
414: This paragraph trails into a a very specific statement rather than resolving the conflict introduced at the start. Please provide a final thought since you are contradicting the findings of the previous paragraph (to an extant).
450: 'Seeing as' is informal language. Rephrase.
454: might provide or even 'might contribute to a better understanding' not 'might assist us in providing'
467: Why are the sample sizes so small here? I feel like I am missing something--are these samples per geographic region discussed previously? If so, remind us again here as the text in the discussion is quite dense and there were several analyses conducted.
476: 'did not, however' or 'however, did not' rather than 'did, however, not'
484: I thought proteins were modelled. Can you not comment on the affect to conformation or structural elements introduced by the mutation?
486: change 'The ACR gene is shown to be functionally" to 'The ACR gene is functionally conserved'
488-490: unclear how Exon 5 is rapidly evolving and there is conservation. The last sentence here is particularly rushed and awkwardly phrased. Please make this text clearer.
497-498: I am not sure what this means. If you are talking about analysis others can do, use the same kind of language used in previous paragraphs.
525, sentence beginning: run-on and hard to follow.
534: nine sites were, not was
535: This final citation does seems misplaced b/c it is part of the earlier discussion placing the results in line with published lit instead of following your findings.
544: Perhaps a sentence or parenthetical note on what those environmental factors are—I seem to remember only rainfall was noted as significant…?
545: All three gene regions—all three genes or three regions in the gene? You didn’t report finding selection in other genes.
548-9: subject-verb agreement: add not adds
551: How specifically will the results aid conservation efforts at reintroduction?
Table 3: use consistent decimalization in each cell, including for 0. I recommend three decimal places for Hd and π
The conclusion needs more detail to summarize specific results, rather than general. Also needed is specific examples of the significance of the work and its specific application to the general areas noted.
In addition to the GLM results requested by the reviewer, I also would like to see a summary of the selection test results.

Reviewer 1 ·

Basic reporting

No comment

Experimental design

No comment

Validity of the findings

No comment

Additional comments

In this paper, which I reviewed previously, the authors examine TLR and ACR diversity in populations of vervet monkeys across southern Africa. Overall the paper is much improved over the previous version. The paper is much improved, and the authors have taken my suggestions on board. I have only two minor comments:

The authors indicate in the Abstract that ACR had no exonic variation [L31], but selection was nevertheless detected [L34]. Selection tests should be undertaken on the coding sequence only (refer http://www.datamonkey.org/help), so I am unclear how selection was detected at the intron.

Paragraph beginning L344: full regression statistics should be provided, perhaps in supplementary material.

Reviewer 2 ·

Basic reporting

.

Experimental design

.

Validity of the findings

.

Additional comments

My comments were properly addressed, I recommend the revised manuscript for publishing.

---

## Round 0.3 · accepted · Accept

Thanks very much for your attention to the last set of revisions that have clarified the remaining manuscript portions that were lacking clarity. I am happy to accept this for publication.

#